# Seroprevalence of Antibodies against Diphtheria, Tetanus and Pertussis in Adult At-Risk Patients

**DOI:** 10.3390/vaccines9010018

**Published:** 2021-01-04

**Authors:** Lise Boey, Eline Bosmans, Liane Braz Ferreira, Nathalie Heyvaert, Melissa Nelen, Lisa Smans, Hanne Tuerlinckx, Mathieu Roelants, Kathleen Claes, Inge Derdelinckx, Wim Janssens, Chantal Mathieu, Johan Van Cleemput, Robin Vos, Isabelle Desombere, Corinne Vandermeulen

**Affiliations:** 1Department of Public Health and Primary Care, Leuven University Vaccinology Center, KU Leuven, Kapucijnenvoer 35, P.O. Box 7001, 3000 Leuven, Belgium; Eline.Bosmans@sfz.be (E.B.); liane.braz.ferreira@gmail.com (L.B.F.); nathalie.heyvaert@telenet.be (N.H.); melissa.nelen@gmail.com (M.N.); lisa.smans@zna.be (L.S.); hanne.tuerlinckx@uzleuven.be (H.T.); corinne.vandermeulen@kuleuven.be (C.V.); 2Environment and Health, Department of Public Health and Primary Care, KU Leuven, Kapucijnenvoer 35, P.O. Box 7001, 3000 Leuven, Belgium; mathieu.roelants@kuleuven.be; 3Department of Nephrology and Renal Transplantation, University Hospitals Leuven, Herestraat 49, 3000 Leuven, Belgium; kathleen.claes@uzleuven.be; 4Department of General Internal Medicine, University Hospitals of Leuven, Herestraat 49, 3000 Leuven, Belgium; inge.derdelinckx@uzleuven.be; 5Department of Respiratory Diseases, University Hospitals of Leuven, Herestraat 49, 3000 Leuven, Belgium; wim.janssens@uzleuven.be; 6Department of Endocrinology, University Hospitals of Leuven, Herestraat 49, 3000 Leuven, Belgium; chantal.mathieu@uzleuven.be; 7Department of Cardiology, University Hospitals of Leuven, Herestraat 49, 3000 Leuven, Belgium; johan.vancleemput@uzleuven.be; 8Department of Respiratory Diseases—Lung Transplantation Unit, University Hospitals of Leuven, Herestraat 49, 3000 Leuven, Belgium; robin.vos@uzleuven.be; 9NRC Bordetella Pertussis and NRC Toxigenic Corynebacteria, Sciensano (Public Health Belgium), Engelandstraat 642, 1180 Ukkel, Belgium; isabelle.desombere@sciensano.be

**Keywords:** seroprevalence, patients, diphtheria, tetanus, pertussis

## Abstract

Patients with chronic diseases are at increased risk of complications following infection. It remains, however, unknown to what extend they are protected against vaccine-preventable diseases. We assessed seroprevalence of antibodies against diphtheria, tetanus and pertussis to evaluate whether current vaccination programs in Belgium are adequate. Antibody titers were assessed with a bead-based multiplex assay in serum of 1052 adults with chronic diseases. We included patients with diabetes mellitus type 1 (DM1) (*n* = 172), DM2 (*n* = 77), chronic kidney disease (*n* = 130), chronic obstructive pulmonary disease (COPD) (*n* = 170), heart failure (*n* = 77), HIV (*n* = 196) and solid organ transplant (SOT) recipients (*n* = 230). Factors associated with seroprevalence were analysed with multiple logistic regression. We found seroprotective titers in 29% for diphtheria (≥0.1 IU/mL), in 83% for tetanus (≥0.1 IU/mL) and 22% had antibodies against pertussis (≥5 IU/mL). Seroprotection rates were higher (*p* < 0.001) when vaccinated within the last ten years. Furthermore, diphtheria seroprotection decreased with age (*p* < 0.001). Tetanus seroprotection was less reached in women (*p* < 0.001) and older age groups (*p* < 0.001). For pertussis, women had more often a titer suggestive of a recent infection or vaccination (≥100 IU/mL, *p* < 0.01). We conclude that except for tetanus, the vast majority of at-risk patients remains susceptible to vaccine-preventable diseases such as diphtheria and pertussis.

## 1. Introduction

Patients with a chronic condition are often at increased risk for complications after exposure to infectious diseases. Although the exact contribution of underlying conditions to infectious disease outcome is not completely elucidated, it is known that patients hospitalised with, for example, severe pertussis often have co-morbidities [1]. As a particular example, patients with chronic pulmonary obstructive disease (COPD) showed to have a 2.5-fold increased risk of being hospitalized due to pertussis [1,2]. In addition, it has been suggested that pathogens such as Bordetella pertussis may lead to exacerbation of diseases. For example, a study in the United States reported a large proportion of patients with COPD exacerbation among patients hospitalized with severe pertussis infection [1]. Hence, patients might end up in a vicious circle where the condition promotes infection and infection worsens the condition [3]. To avoid this, it is imperative that they get vaccinated. The Belgian national immunization technical advisory group (NITAG) advises an adult booster dose with a tetanus and diphtheria containing vaccine every 10 years. This vaccine should include at least once an acellular pertussis component (Tdap vaccine) during adulthood. Whereas childhood vaccination programs mostly meet their intended targets, adult vaccination remains often below the desired coverage level. In the general Belgian adult population, 62% were correctly vaccinated against tetanus in 2008, and diphtheria-tetanus vaccination coverage ranged from 61 to 74% in four other European countries [4,5]. Adult pertussis vaccination coverage is not assessed in Belgium. In high-risk patients, it is often even more challenging to reach a high uptake [6]. These patients are usually followed by a specialist and may therefore visit less often an occupational physician or general practitioner, who is usually in charge of vaccination. Circulating antibodies are, however, needed for protection at the time of exposure to toxins, certainly in the case of tetanus and diphtheria [7,8]. Moreover, protective titers are not always reached in at-risk patients because vaccine immune responses might be impaired [9,10]. Since tetanus is not transmitted from human to human, individual vaccination is the only mode of protection since the principle of herd immunity does not apply. In contrast, vulnerable individuals may benefit from herd immunity when a large proportion of the population is protected against diseases such as diphtheria and pertussis. Unfortunately, relatively few people from the general population have protective titers against these diseases due to waning immunity [11,12,13,14,15]. Moreover, cases of pertussis and diphtheria have resurged in the past few years, albeit more sporadically for diphtheria [14,15,16]. Despite these health risks, serosurveillance studies in the general population and patient groups have been sparse to date. In the present study, we assessed seroprevalence of antibodies against diphtheria, tetanus and pertussis in at-risk patients and factors associated with seroprevalence in a tertiary care hospital in Belgium.

## 2. Materials and Methods

### 2.1. Study Procedure and Population

The present study is a monocentric cross-sectional serosurvey in adult at-risk patients attending the University Hospitals of Leuven. This is a tertiary referral hospital in Belgium, which has about 1800 beds and covers approximately 700,000 outpatient consultations annually [17,18]. All patients older than 18 years who attended the outpatient clinic because of a previous diagnose of diabetes mellitus (DM), heart failure, chronic pulmonary obstructive disease (COPD), chronic kidney disease (CKD), HIV or solid organ transplant (SOT) of lung or heart during a consecutive 5-month recruitment window between September 2014 and March 2016 were invited to the study. Signed informed consent was obtained from all participants. Data were collected with a structured patient interview on vaccination status, disease characteristics and severity and demographic and socio-economic background. Detailed data on the larger survey on vaccination status and determinants of vaccination are reported elsewhere [19]. Disease severity was classified according to international guidelines (the Global Initiative for Chronic Obstructive Lung Disease (GOLD) for COPD, Kidney Disease Improving Global Outcomes (KDIGO) for CKD and the New York Heart Classification (NYHA) for heart failure) [20,21,22]. Severe disease state was defined as KDIGO ≥4, GOLD stage C or D and NYHC classes 3 or 4. Vaccination data were retrieved from documents provided by patients, medical records of the general practitioner or the Flemish vaccination register. Patients were considered correctly vaccinated against diphtheria and tetanus if they received the vaccine within in the last 10 years. Correct pertussis vaccination implied having received at least once a pertussis containing vaccine at adult age or within the past 10 years. The study was performed in accordance with the ethical standards of the Helsinki Declaration and approved by the Ethics Committee Research UZ/KU Leuven of Leuven, Belgium (S56765).

### 2.2. Laboratory Methods

A magnetic bead-based Luminex multiplex assay was used for determination of IgG antibodies against diphtheria toxin (DT), tetanus toxin (TT), pertussis toxin (PT), filamentous hemagglutinin (FHA) and pertactin (Prn) at the Belgian scientific institute of public health (Sciensano) [23]. Anti-DT and anti-TT titers <0.01 IU/mL were considered seronegative and those ≥0.1 IU/mL seroprotective [24]. Anti-PT, anti-FHA and anti-Prn titers ≥5 IU/mL were used as cut-off value for pertussis seropositivity. Since especially anti-PT is related to protection, although there is no correlate of protection, seropositive anti-PT titers were used as indication for pertussis immunity [25]. Anti-PT titers ≥50 IU/mL were indicative for pertussis infection or vaccination in the past 2 years and anti-PT titers ≥100 IU/mL were indicative for a recent infection or vaccination.

### 2.3. Statistical Analysis

Antibody titers below the lower limit of quantification (LLOQ) were replaced by LLOQ divided by two for the calculation of GMTs and confidence intervals. The prevalence rates of seroprotection against tetanus and diphtheria and seroprevalence of pertussis antibodies are reported with exact binomial 95% confidence intervals (95% CI). Determinants (disease type, vaccination status, demographic and socio-economic characteristics) of seroprotection against diphtheria and tetanus and anti-PT seropositivity were analysed with multiple logistic regression. Time since vaccination controlled for age and sex was analysed separately within the group of vaccinated patients. A test probability of 5% was considered statistically significant. All data were analysed with R version 3.0.2 (R Foundation for Statistical Computing, Vienna, Austria, 2013).

## 3. Results

### 3.1. Patient Characteristics

A total of 1331 patients participated in the vaccination coverage study, of whom 1052 (85.5%) gave additional consent for blood sample collection. The present analysis is limited to these patients (66.9% males), whose characteristics are shown in Table 1. A severe disease state was present in all CKD patients, 57.6% of COPD patients and in 41.6% of heart failure patients. In the HIV group, 98.0% had a CD4+ count of ≥200 cells/mm^2^ and 45.4% were men who have sex with men (MSM). Of the 230 SOT patients, 128 patients had received a lung transplantation and 127 a heart transplantation. Overall, less than one third of the patients was vaccinated against diphtheria-tetanus.

### 3.2. Seroprotection and Seroprevalence

The GMTs and the percentage of seroprotective, seropositive, equivocal and seronegative titers are shown in Table 2. Seroprotective titers were reached in 83% of patients for tetanus and in 29% for diphtheria. Furthermore, 36% were seronegative (<0.01 IU/mL) for diphtheria and 2% for tetanus. About half of the patients (46%) had anti-PT antibodies, 8% had anti-PT titers indicative for infection or vaccination in the past few years and 2% had titers indicative for recent infection or vaccination (Table 2). Overall, 13.9% of patients were seroprotected against tetanus and diphtheria and were anti-PT seropositive. Among the different patient groups, CKD patients had the lowest proportion of subjects with protection against tetanus and patients with COPD against diphtheria. Patients with heart failure had the lowest rate of seropositivity for pertussis.

Among the vaccinated patients, 36.6% were protected against diphtheria, 89.9% against tetanus, 67.3% were seropositive for anti-PT, 16.3% had titers indicative of pertussis infection or vaccination in the past few years and 5.1% titers indicative of infection or vaccination in the past few months. Among the incorrectly vaccinated patients, 25.7% were protected against diphtheria, 79.8% against tetanus, 43.7% had an anti-PT titer ≥ 5 IU/mL, 6.6% had titers indicative of infection in the past few years and 1.7% titers indicative of infection in the past few months.

### 3.3. Determinants of Seroprotection and Seroprevalence

The associations between seroprotection against DT and TT or anti-PT seropositivity and vaccination status, disease type, gender and age in adults with chronic diseases are shown in Table 3. For diphtheria, seroprotection increased with recent vaccination and decreased with age. The seroprotection rate was also lower in all disease groups when compared to DM type 1, but this was only statistically significant for COPD and SOT (Table 3). Protective titers for tetanus were more often attained when correctly vaccinated, and less often in woman, and in the oldest age group. The seroprotection rate was significantly lower in all disease groups when compared to DM type 1 (Table 3). As expected for pertussis, vaccinated patients were significantly more often seropositive or more likely to have a titer indicative of an infection or vaccination during the past 2 years. There was a similar trend for titers that indicate a recent infection or vaccination (*p* = 0.06). Women were also more likely to have a titer indicative of recent infection or vaccination (Table 3).

The inclusion of smoking, family income, education and origin in the analyses had a negligible effect on these results (data not shown), except for the effect of vaccination on titers indicative for recent pertussis exposure or vaccination (the OR increases to 3.6; 1.1–10.5; *p* = 0.03). These analyses further revealed that a European origin other than Belgian was associated with better protection against diphtheria (OR vs. Belgian: 2.1; 1.3–3.3; *p* < 0.01). For tetanus, a net family income of more than 3000 euros was associated with better protection (OR vs. 1500–3000 euro: 1.9; 1.0–3.6; *p* < 0,05), and a non-European origin with less protection (OR vs. Belgian: 0.4; 0.2–0.8; *p* < 0.01). For pertussis, past smoking was associated with a seropositive titer (anti-PT ≥ 5 IU/mL) (OR vs. non-smoking: 1.4; 1.1–2.0; *p* = 0.03) and with a titer indicative for previous infection or vaccination (anti-PT ≥ 50 IU/mL) (OR vs. non-smoking:1.8; 1.0–3.5; *p* = 0.05) and active smoking was associated with titers indicative for a recent infection or vaccination (anti-PT ≥ 100IU/mL) (OR vs. non-smoking: 4.5;1.2–18.3; *p* = 0.03).

Within the subgroup of patients who were vaccinated less than 10 years before the study, we did not observe an effect of time since vaccination on protective titers for tetanus (OR: 0.96; 0.84–1.1; *p* = 0.55) or diphtheria (OR: 0.9; 0.9–1.0; *p* = 0.11). For pertussis, the number of years since vaccination was significantly associated with decreased odds for a seropositive anti-PT titer (≥5 IU/mL) (OR: 0.8; 0.7–0.9; *p* = 0.002), past infection or vaccination (anti-PT ≥ 50 IU/mL) (OR: 0.7; 0.4–0.9; *p* = 0.03).

## 4. Discussion

This study demonstrates that a large group of at-risk patients remain susceptible to vaccine-preventable diseases. It provides a comprehensive insight into the seroprotective status of clinical risk groups because of the diversity of clinical conditions and number of patients that were included.

We found a seroprotective status for tetanus in 83% of patients, but only 29% reached protective titers for diphtheria and 46% were anti-PT seropositive. Overall, less than 15% of the patients were protected against tetanus and diphtheria and anti-PT seropositive. These numbers are considerably lower than corresponding data from the general Belgian population [11,12]. In a seroprevalence study from 2006, Theeten et al. reported seroprotective levels for tetanus in more than 90% of persons aged >40 years and for diphtheria in 55% of persons aged 1–65 years [11]. Van der Wielen et al. found a seroprevalence of anti-PT antibodies (≥5 IU/mL) against pertussis in about 70% of those between 1 and 65 years of age in 1993–1994 [12]. Although the comparison might be hampered due to changes in the vaccination programs over the years, the use of different age groups and the lack of vaccination data in population studies, some reasons for these differences can be suggested. A major factor could be the low vaccination uptake in patients as less than 30% were correctly vaccinated against diphtheria and tetanus. In addition, we found that correct vaccination predicts higher seroprevalence rates for all studied diseases.

Time since last vaccination can also influence the level of immunity against vaccine-preventable disease. Evidence exist that antibody titers wane even faster in high-risk groups. Studies in transplant and CKD patients show an accelerated decline, particularly in diphtheria antibodies compared to tetanus antibodies [26,27]. In addition, HIV patients, even those with RNA-HIV below 50 copies/mL, sustain less antibodies due to an impaired cellular immune response [28]. However, we could only find an effect of time since vaccination on anti-PT seropositivity, for which immunity after both vaccination and natural infection is known to be rather short-lived [13]. Therefore, a booster every 10 years might be sufficient to maintain immunity for tetanus and diphtheria, but not for pertussis in this population.

In addition to the influence of vaccination, it is also likely that antibodies were evoked or boosted by natural infection since pertussis has been increasing in Belgium since 2011 [29,30]. Since the pertussis vaccination coverage was less than 10%, we assume that many patients with the high antibody titers had been exposed to wild-type pertussis.

It remains striking, however, that only 38% of the vaccinated patients had protective titers against diphtheria. The exact impact of chronic disease on vaccine immunology is complex, incompletely studied and influenced by many factors, such as the characteristically older age of at-risk patients, comorbidities, disease severity and treatment.

Consistent with previous seroprevalence studies, we found age to be a negative predictor for the seroprevalence of antibodies against diphtheria and tetanus [14,15]. Increased susceptibility might also be related to the immunosuppressive characteristics of chronic disease or the use of immunosuppressive treatment. Rafi et al. showed that increased chronic disease burden may go along with decreased cell-mediated immunity, which in turn might affect humoral immunity [31]. Among the different disease groups in our study, patients with CKD, HIV and SOT had the lowest odds of being protected against tetanus and patients with COPD and SOT the lowest odds for protection against diphtheria. Among SOT patients, the induction and maintenance of antibodies upon vaccination might be reduced due to the immunosuppressive treatment they take to avoid graft rejection [32]. In COPD and CKD patients, vaccine immunology can be impaired by disrupted innate and adaptive immune responses caused by chronic inflammation of the airways and uremic state, respectively [33,34,35]. For HIV patients, reduced vaccine immune response is mostly seen in those with a low CD4-cell count, a detectable viral load and in those not using anti-retroviral therapy. In our study, on the other hand, nearly all patients had CD4+ counts ≥200 cells/µL.

Finally, we also observed some gender differences. Firstly, men were significantly better protected against tetanus than women. This difference has been reported in other seroprevalence studies and has been linked to vaccination practice during the military service and to vaccination after manual work-related injuries [11,12,13,14].

We also looked at titers suggestive of recent pertussis infection or vaccination. In total, 8% had a PT-IgG titer suggestive of an infection or vaccination in the past few years and 2% of a more recent infection or vaccination. Patients with COPD were more likely to have had a recent infection or vaccination compared to DM type 1 patients. This is consistent with another study where the seroprevalence of anti-PT was higher in COPD patients compared to healthy controls [36]. In addition, (past) smoking was associated with pertussis infection or vaccination. Given the low vaccination uptake, this enforces the assumption that both COPD patients and smokers are predisposed to develop respiratory infections such as pertussis due to a reduction of protective functions in the airway epithelium [3,4,5,6,7,8,9,10,11,12,13,14,15,16,17,18,19,20,21,22,23,24,25,26,27,28,29,30,31,32,33,34,35,36,37]. Recent pertussis infections or vaccination were also mainly seen in women. This might be related to vaccination during pregnancy or to the predominant occurrence of pertussis in women, as seen in the general population [30].

Given the high susceptibility of at-risk patients, we advocate for a close follow-up of their vaccination status. Vaccination is the best available tool to prevent infectious diseases, even when vaccine immunity is reduced due to age, disease or treatment. It still has the benefit of reducing the likelihood of acquiring severe disease, such as pertussis infection requiring hospitalization or resulting in post-tussive vomiting [1,2,3,4,5,6,7,8,9,10,11,12,13,14,15,16,17,18,19,20,21,22,23,24,25,26,27,28,29,30,31,32,33,34,35,36,37,38]. Since patients with chronic disease are often followed by a specialist, an appropriate recommendation by the treating specialist may convince them to get their vaccine with their general practitioner. In addition, it remains equally important to avoid transmission of infectious pathogens to vulnerable patients by vaccinating their close contacts and by implementing universal vaccination programs.

Some limitations of our study should be mentioned. The study was performed in a single, albeit large, tertiary care hospital, which may limit extrapolation of the results to all at-risk patients. A general limitation inherent to all seroprevalence studies is the unknown origin of anti-pertussis antibodies, i.e., from vaccination or from natural infection. However, the vaccination coverage was below 10%. We believe that lack of documentation plays a limited role since adult booster vaccination were only recommended since 2013 and cocoon vaccination since 2009, but compliance with these strategies was rather low before and during the recruitment of patients [39]. Unfortunately, we could not include a lifetime history of vaccination (or exposure) as these data could not be reliably collected. Finally, the cut-off values for recent pertussis infection were applied in accordance with international agreements, but patients with chronic diseases may mount less antibodies due to their disease or immunosuppressive therapy.

## 5. Conclusions

In conclusion, our data show that patients with chronic diseases are at increased risk for vaccine preventable diseases. Noticeably, 17% of patients remain susceptible to tetanus, 71% to diphtheria and 54% have low titers to pertussis. This might be explained by the low vaccination coverage, age or the influence of disease and therapy on vaccine immunity. We recommend close follow-up of the vaccination status in patients with chronic diseases and advocate additionally indirect protection through universal vaccination programs and vaccination of their direct contacts.

## Figures and Tables

**Table 1 vaccines-09-00018-t001:** Characteristics of study participants.

	All Patients (*n* = 1052)	DM Type 1 (*n* = 172)	DM Type 2 (*n* = 77)	CKD (*n* = 130)	COPD ^a^ (*n* = 170)	Heart Failure ^b^ (*n* = 77)	HIV (*n* = 196)	SOT (*n* = 230)
**Personal Data**	*n* (%)	*n* (%)	*n* (%)	*n* (%)	*n* (%)	*n* (%)	*n* (%)	*n* (%)
Female gender	348 (33.1)	80 (46.5)	25 (32.5)	43 (33.1)	54 (31.8)	20 (26.0)	53 (27.0)	73 (31.7)
Median age, years (range)	59 (18–92)	44 (18–83)	67 (31–89)	73 (21–91)	65 (29−89)	70 (32–89)	46 (18–75)	59 (19–87)
Age								
<40 years	173 (16.4)	66 (38.4)	2 (2.6)	5 (3.8)	1 (0.6)	2 (2.6)	62 (31.6)	35 (15.2)
40–64 years	492 (46.8)	81 (47.1)	30 (39.0)	25 (19.2)	81 (47.6)	25 (32.5)	125 (63.8)	125 (54.3)
≥65 years	387 (36.8)	25 (14.5)	45 (58.4)	100 (76.9)	88 (51.8)	50 (64.9)	9 (4.6)	70 (30.4)
Smoking								
Smoker	166 (15.8)	30 (17.4)	15 (19.5)	16 (12.3)	30 (17.6)	7 (9.1)	55 (28.1)	13 (5.7)
Ex-smoker	471 (44.8)	47 (27.3)	31 (40.3)	58 (44.6)	132 (77.6)	42 (54.5)	45 (23.0)	116 (50.4)
Net family income								
<1500 euro	225 (21.4)	23 (13.4)	8 (10.4)	30 (23.1)	48 (28.2)	21 (27.3)	52 (26.5)	43 (18.7)
1500–3000 euro	506 (48.1)	85 (49.4)	60 (77.9)	55 (42.3)	83 (48.8)	41 (53.2)	70 (35.7)	112 (48.7)
>3000 euro	217 (20.6)	63 (36.6)	7 (9.1)	15 (11.5)	14 (8.2)	9 (11.7)	64 (32.7)	45 (19.6)
Unknown income	104 (9.9)	1 (0.6)	2 (2.6)	30 (23.1)	25 (14.7)	6 (7.8)	10 (5.1)	30 (13.0)
Educational degree ^c^ (years of study)								
Lower education (<12 years)	355 (33.7)	20 (11.6)	38 (49.4)	64 (49.2)	74 (43.5)	43 (55.8)	45 (23.0)	71 (30.9)
Secondary education (12 years)	369 (35.1)	73 (42.4)	25 (32.5)	40 (30.8)	58 (34.1)	19 (24.7)	74 (37.8)	80 (34.8)
Higher education (>12 years)	320 (30.4)	79 (45.9)	14 (18.2)	24 (18.5)	38 (22.4)	15 (19.5)	75 (38.3)	75 (32.6)
Unknown education	8 (0.8)	0 (0)	0 (0)	2 (1.5)	0 (0)	0 (0)	2 (1.0)	4 (1.7)
Origin ^d^								
Belgian	879 (83.6)	150 (87.2)	68 (88.3)	125 (96.2)	155 (91.2)	71 (92.2)	113 (57.7)	197 (85.7)
European	93 (8.8)	14 (8.1)	5 (6.5)	5 (3.8)	14 (8.2)	3 (3.9)	19 (9.7)	33 (14.3)
Non-European	80 (7.6)	8 (4.7)	4 (5.2)	0 (0)	1 (0.6)	3 (3.9)	64 (32.7)	0 (0)
**Disease data**	*n* (%)	*n* (%)	*n* (%)	*n* (%)	*n* (%)	*n* (%)	*n* (%)	*n* (%)
Relevant comorbid disease ^e^	205 (19.5)	9 (5.2)	6 (7.8)	43 (33.1)	34 (20.0)	24 (31.2)	19 (9.7)	70 (30.4)
Years since diagnosis/transplantation (median (range))	8 (0–64)	18 (0–59)	13 (0–64)	4 (1–47)	7 (0–39)	6 (0–51)	8 (0–30)	7 (1–29)
**Vaccination status**	% (95% CI)	% (95% CI)	% (95% CI)	% (95% CI)	% (95% CI)	% (95% CI)	% (95% CI)	% (95% CI)
Diphtheria-tetanus in the past 10 years	29.1 (26.4–32.0)	26.2 (19.9–33.5)	29.9 (20.2–41.5)	23.1 (16.3–31.4)	34.1 (27.1–41.8)	37.7 (27.1–49.5)	30.6 (24.3–37.7)	26.5 (21.0–32.8)
Any reported pertussis vaccine	9.3 (7.7–11.3)	12.8 (8.4–18.9)	10.4 (4.9–20.0)	4.6 (1.9–10.2)	10.6 (6.6–16.5)	14.3 (7.7–24.5)	3.6 (1.6–7.5)	11.3 (7.7–16.3)

^a^ Patients were classified in categories of disease severity according to Global Initiative for Chronic Obstructive Lung Disease (GOLD) stages: 20.0% had GOLD stage A, 22.4% GOLD stage B, 9.4% GOLD stage C and 48.2% GOLD stage D. The severity of symptoms is measured with the Modified Medical Research Council Dyspnea Scale (mMRC) and the COPD Assessment Test (CAT). Patients with GOLD A and B are at low risk (0–1 exacerbation per year, not requiring hospitalization), GOLD C and D are high risk patients (≥2 exacerbations per year, or one or more requiring hospitalization). GOLD A and C have few symptoms (mMRC 0–1 or CAT <10), GOLD B and D have more symptoms (mMRC ≥2 or CAT ≥10) [20]. ^b^ Patients were classified in categories of disease severity according to New York Heart Classification (NYHA): 26.0% had class I (no limitation in ordinary physical activity), 32.5% class II (mild symptoms and slight limitation during ordinary activity and comfortable at rest), 40.3% had class III (marked limitation in activity due to symptoms, even during less-than-ordinary activity and comfortable only at rest) and 1.3% had class IV (severe limitations and experiences symptoms even while at rest) [21]). ^c^ Education: Lower Education = no secondary school diploma, Secondary education = secondary school diploma achieved, Higher education = university or university college diploma achieved. ^d^ European = At least one of the parents from European geographical area but not from Belgium, Non-European = At least one of the parents was not from the European geographical area. ^e^ Relevant comorbidity is defined as having a comorbid disease that might influence vaccine-induced immunity (metabolic disease, systemic disease immunodeficiencies, renal disease). CKD: chronic kidney disease, COPD: chronic obstructive pulmonary disease, DM: diabetes mellitus, SOT: solid organ transplantation.

**Table 2 vaccines-09-00018-t002:** Geometric mean titers (GMTs) and seroprevalence of antibodies against tetanus toxin, diphtheria toxin, pertussis toxin, pertactin and filamentous hemagglutinin.

	Reference	All Patients (*n* = 1052)	DM Type 1 (*n* = 172)	DM Type 2 (*n* = 77)	CKD (*n* = 130)	COPD (*n* = 170)	Heart Failure (*n* = 77)	HIV (*n* = 196)	SOT (*n* = 230)
**GMT (IU/mL) (95% CI)**									
Anti-DT	-	0.01 (0.01–0.02)	0.04 (0.03–0.06)	0.01 (0.00–0.01)	0.01 (0.01–0.01)	0.01 (0.00–0.01)	0.01 (0.01–0.02)	0.04 (0.03–0.05)	0.01 (0.01–0.01)
Anti-TT	-	0.54 (0.49–0.61)	1.38 (1.13–1.67)	0.40 (0.27–0.58)	0.31 (0.22–0.43)	0.48 (0.37–0.62)	0.57 (0.40–0.83)	0.52 (0.41–0.65)	0.47 (0.37–0.60)
Anti-PT	-	4.21 (3.83–4.64)	3.97 (3.09–5.10)	5.33 (3.80–7.48)	5.13 (3.89–6.78)	4.52 (3.53–5.79)	3.72 (2.71–5.09)	3.57 (2.87–4.44)	4.14 (3.38–5.07)
Anti-FHA	-	22.7 (21.0–24.5)	22.6 (19.0–26.9)	29.4 (22.9–37.8)	32.1 (26.3–39.5)	26.3 (22.2–31.2)	37.3 (29.3–47.6)	15.6 (12.9–18.9)	17.9 (14.9–21.5)
Anti-Prn	-	9.94 (9.00–11.0)	19.1 (14.7–24.9)	7.90 (5.70–11.0)	6.81 (5.18–8.95)	9.86 (7.96–12.6)	14.5 (9.67–21.8)	8.58 (6.95–10.6)	8.20 (6.70–10.0)
**Seroprotection**		% (95% CI)	% (95% CI)	% (95% CI)	% (95% CI)	% (95% CI)	% (95% CI)	% (95% CI)	% (95% CI)
Diphtheria	≥0.1 IU/mL	28.9 (26.2–31.7)	45.9 (38.3–53.7)	22.1 (13.4–33.0)	21.5 (14.8–29.6)	17.1 (11.7–23.6)	22.1 (13.4–33.0)	41.3 (34.4–48.6)	23.0 (17.8–29.0)
Tetanus	≥0.1 IU/mL	82.6 (80.2–84.8)	95.3 (91.0–98.0)	79.2 (68.5–87.6)	72.3 (63.8–79.8)	80.0 (73.2–85.7)	83.1 (72.9–90.7)	85.2 (79.4–89.9)	79.6 (73.8–84.6)
**Seronegativity**		% (95% CI)	% (95% CI)	% (95% CI)	% (95% CI)	% (95% CI)	% (95% CI)	% (95% CI)	% (95% CI)
Diphtheria	<0.01 IU/mL	35.6 (32.7–38.6)	20.9 (15.1–27.8)	50.6 (39.0–62.2)	43.8 (35.2–52.8)	48.2 (40.5–56.0)	36.4 (25.7–48.1)	19.4 (14.1–25.6)	41.3 (34.9–48.0)
Tetanus	<0.01 IU/mL	2.4 (1.5–3.5)	0.6 (0.01–3.2)	1.3 (0.03–7.0)	6.2 (2.7–11.8)	2.4 (0.6–5.9)	0.00 (0.00–4.7)	2.0 (0.6–5.1)	3.0 (1.2–6.2)
**Pertussis seroprevalence**		% (95% CI)	% (95% CI)	% (95% CI)	% (95% CI)	% (95% CI)	% (95% CI)	% (95% CI)	% (95% CI)
Anti-PT	≥5 IU/mL	45.9 (42.9–49.0)	44.8 (37.2–52.5)	55.8 (44.1–67.2)	53.1 (44.1–61.9)	47.1 (39.4–54.9)	31.2 (21.1–42.7)	42.9 (35.8–50.1)	46.1 (39.5–52.8)
Anti-FHA	≥5 IU/mL	89.3 (87.2–91.1)	90.1 (84.6–94.1)	93.5 (85.5–97.9)	93.8 (88.2–97.3)	95.3 (90.9–97.9)	100.0 (95.3–100.0)	82.7 (76.6–87.7)	82.2 (76.6–86.9)
Anti-Prn	≥5 IU/mL	64.3 (61.3–67.2)	78.5 (71.6–84.4)	58.4 (46.6–69.6)	57.7 (48.7–66.3)	64.1 (56.4–71.3)	75.3 (64.2–84.4)	61.2 (54.0–68.1)	58.3 (51.6–64.7)
Pertussis infection in last 2 years	≥50 IU/mL	7.5 (6.0–9.3)	7.6 (4.1–12.6)	7.8 (2.9–16.2)	9.2 (4.9–15.6)	9.4 (5.5–14.8)	7.8 (2.9–16.2)	5.6 (2.8–9.8)	6.5 (3.7–10.5)
Recent pertussis infection	≥100 IU/mL	2.0 (1.2–3.0)	1.2 (0.1–4.1)	1.3 (0.03–7.0)	2.3 (0.5–6.6)	3.5 (1.3–7.5)	1.3 (0.03–7.0)	1.5 (0.3–4.4)	2.2 (0.7–5.0)

DT: diphtheria toxin, TT: tetanus toxin, PT: pertussis toxin, FHA: filamentous hemagglutinin, Prn: pertactin, CKD: chronic kidney disease, COPD: chronic obstructive pulmonary disease, DM: diabetes mellitus, SOT: solid organ transplantation, CI: confidence interval.

**Table 3 vaccines-09-00018-t003:** Determinants of seroprevalence for diphtheria, tetanus and of pertussis: multiple logistic regression.

*n* = 1052	Diphtheria (≥0.1 IU/mL)	Tetanus (≥0.1 IU/mL)	Pertussis (≥5 IU/mL)	Pertussis (≥50 IU/mL)	Pertussis (≥100 IU/mL)
	OR (95% CI)	OR (95% CI)	OR (95% CI)	OR (95% CI)	OR (95% CI)
Age					
<40 years	Reference	Reference	Reference	Reference	Reference
40–64 years	0.3 (0.2–0.5) ***	0.7 (0.3–1.2)	0.9 (0.6–1.3)	1.0 (0.5–2.0)	0.8 (0.2–3.1)
≥65 years	0.2 (0.1–0.3) ***	0.2 (0.1–0.4) ***	0.8 (0.5–1.3)	0.6 (0.3–1.4)	0.2 (0.0–1.2) °
(Correctly) vaccinated ^§^	1.8 (1.3–2.4) ***	2.4 (1.6–3.7) ***	2.8 (1.8–4.5) ***	2.7 (1.4–4.8) **	2.9 (0.9–8.0) °
Disease group					
DM type 1	Reference	Reference	Reference	Reference	Reference
DM type 2	0.6 (0.3–1.2)	0.3 (0.1–0.7) **	1.7 (1.0–3.1) °	1.4 (0.5–3.9)	2.7 (0.1–32.2)
CKD	0.7 (0.4–1.3)	0.2 (0.1–0.5) ***	1.6 (1.0–2.7) °	2.0 (0.8–5.0)	7.2 (1.0–65.7) °
COPD	0.4 (0.3–0.7) **	0.3 (0.1–0.6) **	1.2 (0.8–1.9)	1.6 (0.7–3.8)	6.2 (1.2–48.0) *
Heart failure	0.6 (0.3–1.2)	0.4 (0.1–1.0) *	0.6 (0.3–1.0) °	1.3 (0.4–3.8)	2.9 (0.1–34.5)
HIV	0.8 (0.5–1.3)	0.2 (0.1–0.4) ***	1.0 (0.7–1.5)	0.8 (0.4–2.0)	2.0 (0.3–15.9)
SOT	0.5 (0.3–0.7) **	0.2 (0.1–0.4) ***	1.1 (0.7–1.7)	1.0 (0.4–2.2)	3.0 (0.6–21.5)
Female (vs. male)	1.0 (0.7–1.3)	0.3 (0.2–0.5) ***	0.9 (0.7–1.2)	1.2 (0.7–2.0)	3.4 (1.4–8.8) **

CKD: chronic kidney disease, COPD: chronic obstructive pulmonary disease, DM: diabetes mellitus, SOT: solid organ transplantation. ^§^ Correctly vaccinated against diphtheria and tetanus: ≥1 dose within in the last 10 years. Correctly vaccinated against pertussis: ≥1 dose with a pertussis containing vaccine at adult age or within the past 10 years. *p* < 0.1, * *p* < 0.5; ** *p* < 0.01, *** *p* < 0.001. ° *p* < 0.1.

## Data Availability

Data is contained within the article.

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
