# Peer review of "Seroprevalence of Antibodies against Diphtheria, Tetanus and Pertussis in Adult At-Risk Patients"

_vaccines, 2021, doi:10.3390/vaccines9010018_

Round 1

Reviewer 1 Report

The manuscript “Seroprevalence of antibodies against diphtheria, tetanus and pertussis in adult at-risk patients” describes a monocentric cross-sectional serosurvey in adult patients suffering from chronic diseases, to assess their titer of antibodies against tetanus, diphtheria, and pertussis toxins. The rationale of the study is clear, the paper is well written and adequately discussed.

The fact that women have high titer of anti-PT (>100IU/mL) (OR 3.4 for women vs men) can it be ascribed to the fact the women often receive pertussis vaccine during pregnancy? Therefore, women are more likely vaccinated than men? Or is this data statistically aligned with the report of a higher incidence of pertussis infection in women? Reference 38, quoted in line 270 does not discuss pertussis, but tetanus incidence.

Author Response

Dear reviewer,

Thank you very much for giving us the opportunity to submit a revised version of our manuscript. We are also grateful for your comments and suggestions, which have been helpful in improving the manuscript. Below you will find a reply to your comments, which was approved by all authors. All changes in the manuscript are marked with track changes.

Reviewer 1

The manuscript “Seroprevalence of antibodies against diphtheria, tetanus and pertussis in adult at-risk patients” describes a monocentric cross-sectional serosurvey in adult patients suffering from chronic diseases, to assess their titer of antibodies against tetanus, diphtheria, and pertussis toxins. The rationale of the study is clear, the paper is well written and adequately discussed.

The fact that women have high titer of anti-PT (>100IU/mL) (OR 3.4 for women vs men) can it be ascribed to the fact the women often receive pertussis vaccine during pregnancy? Therefore, women are more likely vaccinated than men? Or is this data statistically aligned with the report of a higher incidence of pertussis infection in women? Reference 38, quoted in line 270 does not discuss pertussis, but tetanus incidence.

Answer: Thank you for your useful comment. We assume there are two potential reasons for the higher titer of anti-PT (>100IU/mL) in women. The first reason is indeed vaccination during pregnancy and the second reason might be that women get infected more often due to close contacts with possible pertussis cases, for example because of caregiving. However, our study was not designed to assess the influence of these factors. We elaborated on this point in the discussion and corrected the reference (thank you for that!) on line 270.

Reviewer 2 Report

The study by Lise Boey et al assessed seroprotection against diphtheria, tetanus and seroprevalence of pertussis antibodies in at-risk patients. In particular, they analysed a cohort of 1052 subjects with specific chronic diseases: type 1 and type 2 diabetes mellitus, chronic kidney disease, COPD, heart failure, HIV, and solid organ transplant of lung or heart. The severity of each disease was evaluated with an appropriate score. They found seroprotective titers in 29% for diphteria, in 83% for tetanus, and 22% had antibodies against pertussis. Authors further analyzed factor associated with seroprevalence and found that seroprotection decreased with age for diphteria, while tetanus seroprotection was less reached in women and older age groups. As for pertussis, women had a higher titer as compared to male.

Major

Albeit the epidemiological data are interesting in itself, one should be noted that a number of previous article reported similar results. In addition, results reported by the multivariate regression analysis are mostly confirmatory of previous studies.

Minor

  • Aim reported in the Introduction (see lines 79-81) is not consistent with analyses reported in the Results. Indeed, they also reported factors associated with seroprevalence for each disease.
  • Patients with SOT could be on immunosuppressive drugs. How was managed this variable?

Author Response

Dear reviewer,

Thank you very much for giving us the opportunity to submit a revised version of our manuscript. We are also grateful for your comments and suggestions, which have been helpful in improving the manuscript. Below you will find a point by point reply to these comments, which were approved by all authors. All changes in the manuscript are marked with track changes.

Reviewer 2

The study by Lise Boey et al assessed seroprotection against diphtheria, tetanus and seroprevalence of pertussis antibodies in at-risk patients. In particular, they analysed a cohort of 1052 subjects with specific chronic diseases: type 1 and type 2 diabetes mellitus, chronic kidney disease, COPD, heart failure, HIV, and solid organ transplant of lung or heart. The severity of each disease was evaluated with an appropriate score. They found seroprotective titers in 29% for diphteria, in 83% for tetanus, and 22% had antibodies against pertussis. Authors further analyzed factor associated with seroprevalence and found that seroprotection decreased with age for diphteria, while tetanus seroprotection was less reached in women and older age groups. As for pertussis, women had a higher titer as compared to male.

Major

Albeit the epidemiological data are interesting in itself, one should be noted that a number of previous article reported similar results. In addition, results reported by the multivariate regression analysis are mostly confirmatory of previous studies.

Answer: Thank you for your feedback. We agree that the factors associated with seroprevalence in this study are largely in agreement with what can be found in literature. We have now put some more emphasis on this in the discussion (lines 244 and 259). Nevertheless, our study adds to the literature by simultaneously assessing the seroprevalence of several vaccine preventable diseases in a broad range of at-risk groups who are more susceptible to a severe disease course and for which studies have been sparse. Other studies are usually limited to a single patient group and/or a single vaccine. As such we believe that our manuscript has an added value over limited number of previously published studies.

Minor

  • Aim reported in the Introduction (see lines 79-81) is not consistent with analyses reported in the Results. Indeed, they also reported factors associated with seroprevalence for each disease.

Answer: Thank you for your useful comment. We reformulated the aim at line 79-81 as follows: “In the present study we assessed seroprevalence of diphtheria, tetanus and pertussis in at-risk patients and factors associated with seroprevalence in a tertiary care hospital in Belgium.”

  • Patients with SOT could be on immunosuppressive drugs. How was managed this variable?

Answer: All SOT patients in our study used immunosuppressive medication, which is why we considered them as the most immunocompromised group in the study. In order not to make this manuscript unnecessarily difficult, we did not report within-group differences. We agree, however, that the degree of immunosuppression in the SOT group could be variable. Although we did not assess which particular immunosuppressive drugs the patients used, we know that lung transplant patients in our hospital often use a combination of three immunosuppressive agents. For this reason, they are often more severely immunocompromised than heart transplant patients. However, when comparing the seroprevalence rates in heart and lung transplant patients, and controlling for age, vaccination status and sex, there is only a statistically significant difference for seroprotection against diphtheria. Since we did not observe a significant effect for the other diseases and for the sake of clarity of our paper, we prefer not to add this information. However, if the editor or reviewer believe this is of added value, we will be very happy to include the results of this analysis.

Round 2

Reviewer 2 Report

The Authors replied to the minor revisions. I agree that the article is well written, but my major concern is the novelty.